# Prediction of Landslide Displacement Based on the Variational Mode Decomposition and GWO-SVR Model

Chenhui Wang [1],* and Wei Guo [2]

1 School of Automation Engineering, University of Electronic Science and Technology of China, Chengdu 611731, China
2 Center for Hydrogeology and Environmental Geology Survey, China Geological Survey, Baoding 071051, China
* Correspondence: wangchenhui@mail.cgs.gov.cn

**Abstract:** Accurate prediction of landslide displacement is an effective way to reduce the risk of landslide disaster. Under the influence of periodic precipitation and reservoir water level, many landslides in the Three Gorges Reservoir area underwent significant displacement deformation, showing a similar step-like deformation curve. Given the nonlinear characteristics of landslide displacement, a prediction model is established in this study according to the variational mode decomposition (VMD) and support vector regression (SVR) optimized by gray wolf optimizer (GWO-SVR). First, the original data are decomposed into trend, periodic and random components by VMD. Then, appropriate influential factors are selected using the grey relational degree analysis (GRDA) method for constructing the input training data set. Finally, the sum of the three displacement components is superimposed as the total displacement of the landslide, and the feasibility of the model is subsequently tested. Taking the Shuizhuyuan landslide in the Three Gorges Reservoir area as an example, the accuracy of the model is verified using the long time-series monitoring data. The results indicate that the newly proposed model achieves a relatively good prediction accuracy with data decomposition and parameter optimization. Therefore, this model can be used for the predict the accuracy of names and affiliations ion of landslide displacement in the Three Gorges Reservoir area.

**Keywords:** displacement prediction; variational mode decomposition; grey wolf optimizer; support vector regression

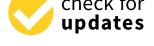



## 1. Introduction

Landslides are one of the most severe types of geological hazards in nature [1]. Landslide hazards often cause great losses, including property damage, injury and loss of life [2,3]. Due to the complex geological conditions and frequent tectonic activities in China, landslide disasters are frequent and may lead to catastrophic damage [3,4]. Displacement prediction is intuitive and important for real-time monitoring and early warning of landslides. Accurate prediction of landslide displacement can reduce the risk, and has become an increasingly important research issue in recent years [5,6]. Many researchers are committed to the prediction of landslide displacement [7–9]. Early landslide prediction is mainly empirical, which uses certain macroscopic signs of landslides to speculate on the time of occurrence. With the continuous improvement of the landslide monitoring technology, more and more means are available to obtain landslide monitoring data [10–13]. Growing attention has been paid to landslide displacement prediction from the background of landslide conception and genesis mechanism, by combining landslide multi-source sensing information, displacement prediction-based mathematical statistical analysis, non-linear prediction and comprehensive coupling model, which can provide more complex analytical ideas [3,14–16]. The empirical model requires creep experiments to validate the prediction model and has limited application scenarios. Mathematical statistical models are better for

the prediction of single influential factors, but cannot solve the displacement prediction of multiple influential factors. Nonlinear prediction suffers from slow convergence and is easily trapped in local minima. The integrated coupled model achieves the prediction of landslide displacement from multiple model perspectives and improves the accuracy of displacement prediction. It has been reported that the obvious step-like displacement can be affected by the periodic fluctuation of special geological environments, seasonal rainfall and reservoir water level adjustment [5,16,17]. Landslide deformation is mainly caused by a variety of factors. The complex process of landslide deformation makes it difficult to accurately distinguish the stages of landslide deformation [18–21]. Therefore, the traditional empirical means are no longer applicable.

At present, nonlinear integrated models based on long time-series analysis is the most common displacement prediction method [22–24]. Many studies have classified the original landslide displacement data into different characteristic components by analyzing the evolution mechanism of landslide deformation. These methods consider both external and internal factors that induce landslide deformation to achieve satisfactory prediction results [17]. The conventional decomposition methods mainly include moving average method, fitting a polynomial trend and smoothing a priori method, which mainly extract trend displacement and periodic displacement [8]. However, these methods do not fully consider the influence of random factors and failed to obtain the random displacement. More comparisons are needed to determine the decomposition parameters, and the computational efficiency is relatively low [25]. In addition, researchers attempt to obtain different components of landslide displacement using the empirical mode decomposition (EMD) [26,27], the ensemble EMD (EEMD) [18,23,28], the complete EEMD of adaptive noise (CEEMDAN) [29,30] and wavelet transform [31]. The above methods can overcome the shortcomings of the conventional decomposition methods, and can completely decompose the displacement components with different characteristics. However, a fixed displacement component cannot be obtained by these methods. The EMD achieves a thorough decomposition of the original displacement of the landslide, but it also suffers from modal confounding and computational inefficiency [29]. In recent years, several researchers have used variational mode decomposition (VMD) combined with artificial intelligence algorithms to achieve accurate prediction of landslide displacement [32,33]. The VMD can select the number of features of displacement decomposition according to the data size and type, which can determine the physical significance of each feature component. It has higher decomposition efficiency and accuracy than the conventional methods, including EMD and EEMD. Therefore, the original landslide displacement data are effectively extracted by VMD to obtain the most optimal landslide displacement component [34].

In recent years, artificial intelligence algorithms, such as the back propagation neural network (BPNN), Support Vector Regression model (SVR) and Elman neural network model, have been increasingly used for the prediction of landslide displacement [35,36]. For complex nonlinear curves, BPNN converges slowly and may not achieve satisfactory fitting performance. The weights and threshold inputs of the Elman neural network have a random nature and result in reduced model prediction accuracy [36]. Classical machine learning algorithms often have difficulty in determining the parameters of the optimization model. SVR has great flexibility in dealing with nonlinear data and helps to solve the nonlinear regression problem [37]. However, it suffers from a shortage of parameter selection, which requires appropriate parameter optimization methods to solve the problem and improve the prediction accuracy [38]. Numerous parameter search methods have been proposed by researchers in recent years, including the grid search method, genetic algorithm (GA) [39,40] and particle swarm optimization (PSO) [5,27,35,41]. The gray wolf optimizer (GWO) has been widely used in combinatorial model optimization problems compared to some existing algorithms (GA, PSO), and in particular, it can greatly improve the efficiency of parameter optimization. Therefore, the GWO algorithm has also been introduced to realize the optimization of the SVR model parameters [25,42].

The aim of this study is to establish a novel landslide displacement prediction method based on the VMD and GWO-SVR model. Taking the Shuizhuyuan landslide in the Three Gorges Reservoir area as an example, the original landslide displacement data are decomposed by the VMD method into trend, periodic and random components. The external and internal influence factors of landslide displacement are selected by combining the gray relational degree analysis (GRDA). Then, the data are divided into training and validation sets with different time scales. The optimal combination model is established using the GWO-SVR model, which can achieve displacement prediction for the test set. Finally, the effectiveness of the model is analyzed.

## 2. Methodology

### 2.1. Variational Mode Decomposition

VMD is an effective method for mode variational problems and signal processing [34]. The VMD algorithm can effectively extract the characteristic information of nonlinear and non-stationary signals. The algorithm is mainly used for constructing and solving variational problems. The basic principle of VMD is to decompose the original $f(t)$ signal into $K$ characteristic components when the sum of the bandwidths of each mode component ($u_k$) is minimum. The VMD selects the appropriate number of mode decompositions based on the original signal and obtains the estimated fundamental frequency bandwidth and converts the problem for solving a variational problem with constraints [43].

$$\begin{cases} \min\limits_{u_k, \omega_k} \left\{ \sum\limits_{k=1}^{K} \left\| \partial_t \left[ \left( \sigma(t) + \frac{j}{\pi t} \right) u_k(t) \right] e^{-j\omega_k t} \right\|_2^2 \right\} \\ s.t. \sum\limits_{k=1}^{K} u_k = f(t) \end{cases} \tag{1}$$

where $\omega_k$ is the center frequency; $(\sigma(t) + j/\pi t)u_k(t)$ is the analytical signal. Then, the Lagrange multiplier operator $\lambda$ and the quadratic penalty term $\alpha$ are introduced to transform the above problems into unconstrained problems. The saddle points are obtained by continuously updating $\lambda^{n+1}$, $\omega_k^{n+1}$ and $u_k^{n+1}$ using the alternating direction method of multipliers. Decomposition of the initial signal $f(t)$ into $K$ characteristic components is then conducted.

### 2.2. Support Vector Regression Model

SVR is a method for solving nonlinear function regression problems, which has strong sparsity and robustness [37]. SVR mainly uses kernel functions to transform low-dimensional nonlinearities into higher-level linear problems. After the solution, it is mapped back to the low-dimensional space. The primary operations of SVR model are described as follows.

Given a training sample $D$, a regression model $f(x)$ is learned approximate to $y$.

$$D = \{(x_1, y_1), (x_2, y_2), \cdots, (x_m, y_m)\}, y_i \in R \tag{2}$$

$$f(x) = w^T + b \tag{3}$$

where $x_i = \{x_{i1}, x_{i2}, \cdots, x_{ij}\}$ is the input vector, $y_i$ is the corresponding output vector, and $w$ and $b$ are the measured parameters. In this model, the loss is zero only when $y$ and $f(x)$ are identical. It can be assumed by the SVR that a deviation of $\varepsilon$ is tolerable between $y$ and $f(x)$. The loss is measured only when the absolute value of the difference between $f(x)$ and $y$ is greater than $\varepsilon$, which corresponds to the construction of an interval band of width $2\varepsilon$ centered on $f(x)$. If the training samples fall into this interval band, they are deemed to be correctly predicted. The kernel function in this model adopts radial basis function [30].

### 2.3. Grey Wolf Optimizer

GWO is a meta-heuristic intelligent algorithm, which has the characteristics of simple structure, few parameters and fast convergence [42]. The main contents of GWO algorithm are as follows [30,44].

① Social rank. Grey wolves are divided into four grades according to their fitness value.

② Surrounding prey. Grey wolves need to encircle their prey, and the position of individual wolves is constantly updated [45].

$$\vec{D} = \left| \vec{C} \cdot \vec{X}_{\mathrm{p}}(t) - \vec{X}(t) \right| \tag{4}$$

$$\vec{X}(t+1) = \vec{X}_{\mathrm{p}}(t) - \vec{A} \cdot \vec{D} \tag{5}$$

where $\vec{D}$ is the distance between the individual gray wolf and the prey; $\vec{X}_{\mathrm{p}}(t)$ is the global optimal solution position; $\vec{X}(t)$ is the position of the gray wolf; $t$ is the number of current iterations; $\vec{C}$ and $\vec{A}$ is the coefficient vector.

③ Hunting. After the gray wolves surrounded the prey, they started to hunt, and generally updated their respective positions at $\alpha$, $\beta$ and $\delta$.

$$\vec{D}_k = \left| \vec{C}_i \cdot \vec{X}_k(t) - \vec{X}(t) \right| \tag{6}$$

$$\vec{X}_i = \vec{X}_k - \vec{A}_i \times \vec{D}_k \tag{7}$$

$$\vec{X}_i(t+1) = \frac{\vec{X}_1 + \vec{X}_2 + \vec{X}_3}{3} \tag{8}$$

where $\vec{D}_k$ is $\left\{ \vec{D}_\alpha, \vec{D}_\beta, \vec{D}_\delta \right\}$, the distances of $\alpha$, $\beta$ and $\delta$ from $\omega$, respectively, and $\vec{X}(t+1)$ is the location of gray wolf following each update.

④Find the best adaptation of gray wolf $\vec{X}_\alpha$ and output the optimal parameters $g$ and $C$.

### 2.4. Displacement Prediction Flow

Figure 1 illustrates the landslide displacement process of this study.

Data decomposition. The original displacement is decomposed into trend, period and random components by VMD. The influential factors are decomposed into periodic and random components.

(1) Integration of the data set. The data obtained from the decomposition are employed to construct separate datasets for the three components.
(2) Model training. The GWO-SVR model is trained to achieve the optimal prediction models for the respective components.
(3) Selection of the optimal model. The model of three components is superimposed to obtain the cumulative displacement optimal prediction model.
(4) Analysis of the results of the optimal model. The model is verified in the test set, and the accuracy is determined in combination with each evaluation indicator.

To quantitatively analyze the forecast accuracy, the mean absolute error (MAE), root mean square error (RMSE) and coefficient of determination ($R^2$) are chosen as the evaluation indicators of the model performance. The evaluation indexes are as follows.

$$\mathrm{MAE} = \frac{1}{N} \sum_{i=1}^{N} |\hat{x}_i - x_i| \tag{9}$$

$$\text{RMSE} = \sqrt{\frac{1}{N}\sum_{i=1}^{N}(\hat{x}_i - x_i)^2} \qquad (10)$$

$$\text{R}^2 = 1 - \frac{\sum\limits_{i=1}^{N}(\hat{x}_i - x_i)^2}{\sum\limits_{i=1}^{N}(\overline{x}_i - x_i)^2} \qquad (11)$$

where $N$ is the number of samples, $\hat{x}_i$ and $x_i$ are the predicted and actual values, respectively, and $\overline{x}_i$ is the average of the actual value. The prediction results with a smaller RMSE and larger $\text{R}^2$ are considered to be better.

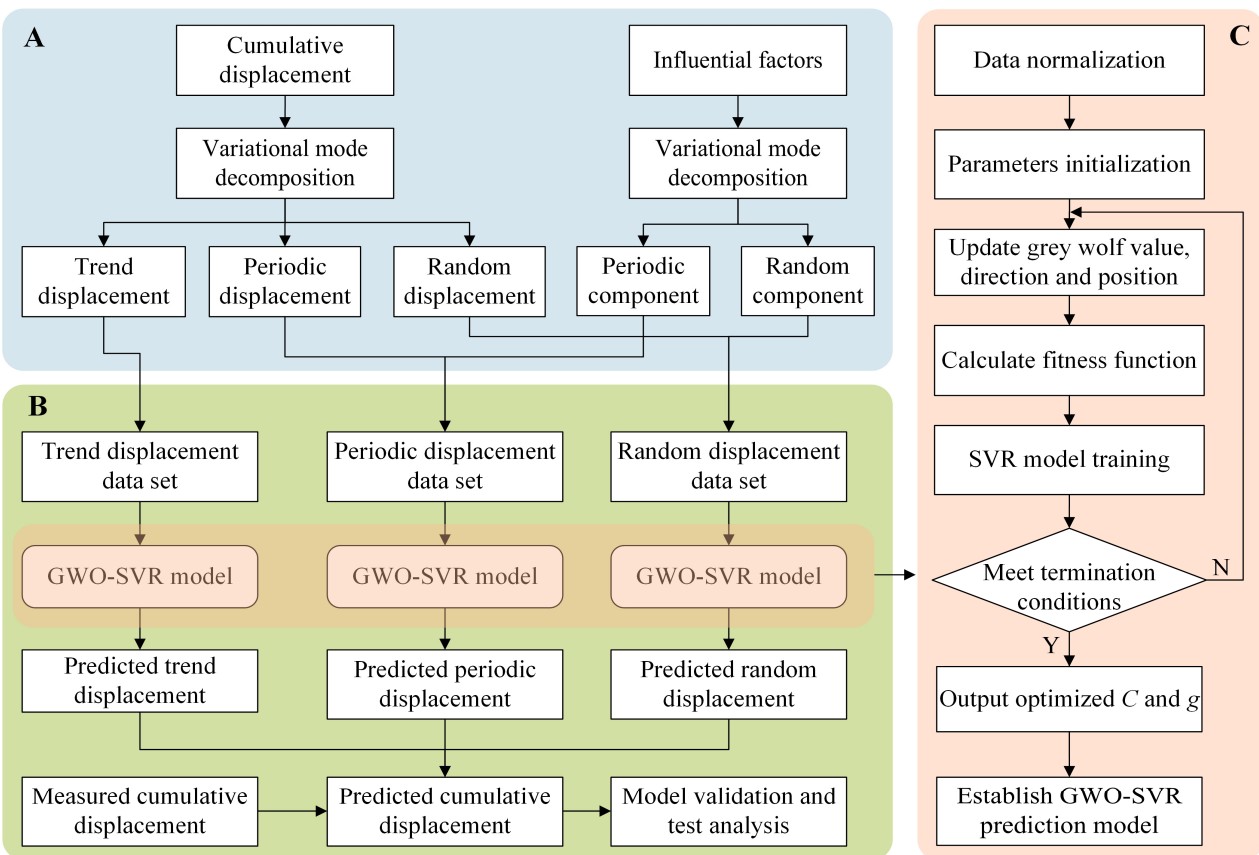

**Figure 1.** Flowchart of displacement prediction. (**A**) Data preprocessing (light blue); (**B**) model training, validation and test analysis (light green); (**C**) establishment of GWO-SVR prediction model (light orange).

## 3. Case Study: Shuizhuyuan Landslide

### 3.1. Geological Conditions

The Shuizhuyuan landslide is 14.82 km east of Wushan city and 170 km from the Three Gorges Dam [43]. The longitudinal length of the landslide is about 800 m; the horizontal width ranges from 360 m to 1200 m; the average thickness is 30 m; the area is $62 \times 10^4$ m$^2$; and the volume is $1850 \times 10^4$ m$^3$. The section with elevation below 180~250 m has a relatively large topographic gradient of 30~45°, which is the front edge area of the landslide. At the elevation of 180~250 m and below 300 m, it is mainly gentle slope, which is the central and rear-edge area of the landslide (Figure 2).

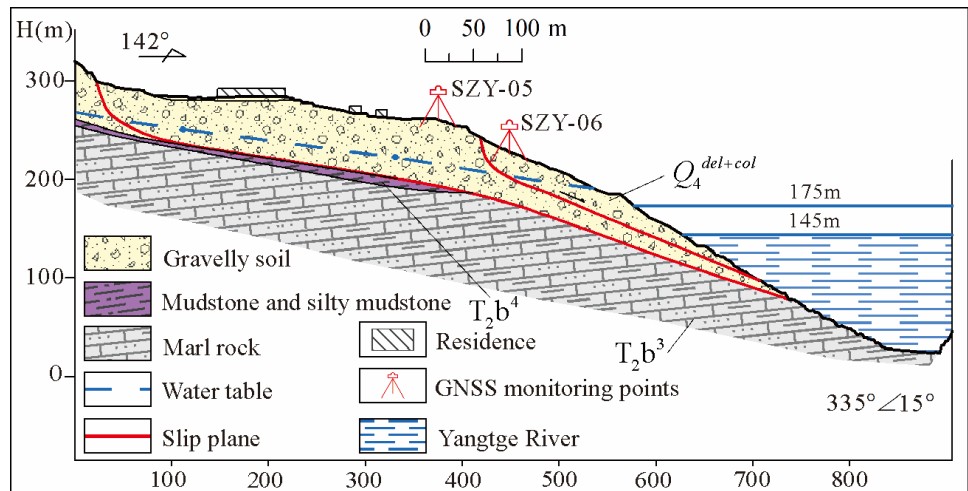

**Figure 2.** Geological profile of Shuizhuyuan landslide.

The landslide material is mainly composed of Quaternary landslide accumulation gravelly soil layer with loose structure, which is easy to be softened by rainfall infiltration and causes sliding deformation. The contact surface between the surface loose soil and bedrock is the sliding surface of the landslide. Rainfall infiltration and reservoir water level change can cause the hydraulic gradient of groundwater in the landslide body to increase, and the pore water pressure to increase, which will increase the sliding force of the landslide. In recent years, the landslide has generally presented a trend of nearly uniform creep deformation, which is mainly characterized by large deformation at the front edge and relatively small deformation at the rear edge.

### 3.2. Analysis of Deformation Characteristics

The landslide mainly includes GNSS surface displacement and rainfall monitoring. Combined with the landslide deformation trend, the monitoring point SZY-03 is located in the front edge of the landslide and its data are the most obvious. Therefore, SZY-03 monitoring point among them is selected for data analysis, as shown in Figure 3.

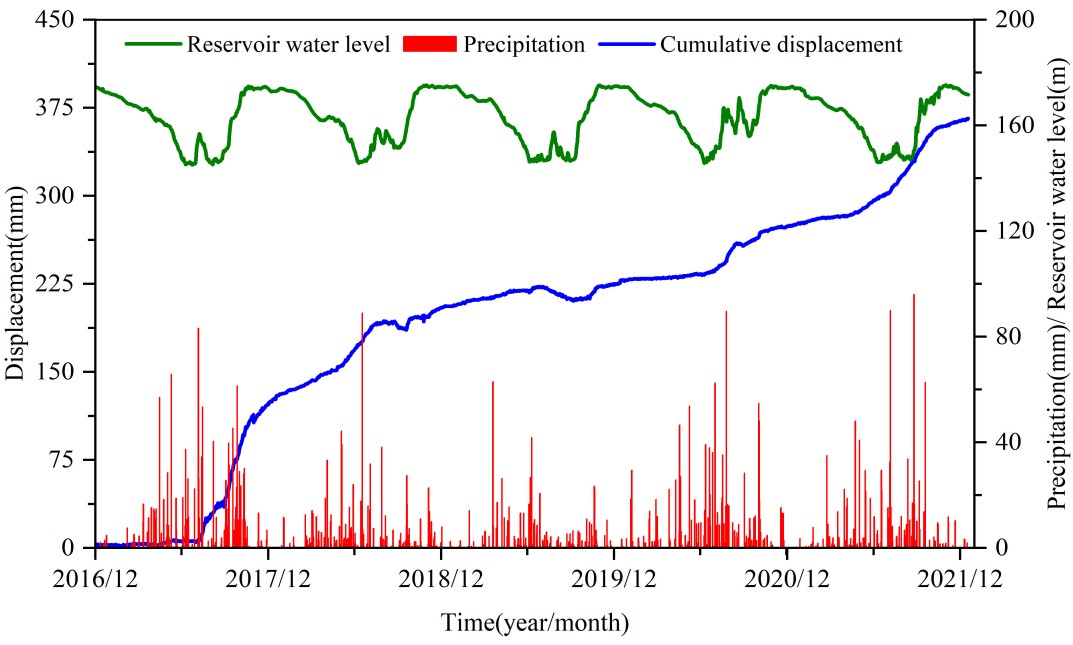

**Figure 3.** Monitoring data of the Shuizhuyuan landslide.

The landslide is affected by the periodic rise and fall of the reservoir water level as well as the annual rainy season. During the adjustment of the reservoir water level and abundant precipitation, the displacement deformation is faster, while the deformation is slow during the non-rainy season or the relatively stable period of the reservoir water level. Therefore, the displacement has relatively obvious step-like characteristics (Figure 3). The displacement shows an upward trend when the abundant precipitation is concentrated from May to September every year. For example, the precipitations in 2017 were 916.6 mm and 445.5 mm in the rainy and non-rainy seasons, with the average monthly variations of the corresponding displacement of 11.2 mm/month and 7.1 mm/month, respectively. From November 2017 to June 2019, the reservoir water level fluctuated within the range of 147–174 m. The displacement expanded from 5.7 mm to 220.9 mm during the 2017 rainy season, and the water level was reduced to 147 m. During the annual rainy season and reservoir water level adjustment cycle, the landslide displacement changes significantly. Precipitation contributes to the softening of the slope structure and makes the landslides prone to displacement. The falling water level weakens the support force of the landslide body, which in turn leads to further increase of the landslide displacement. On the contrary, the rising water level can effectively support and protect the landslide body, and the displacement is relatively stable. From January 2020 to December 2021, the reservoir water levels fluctuated periodically in the range of 147–174 m, and precipitation was mainly concentrated in the rainy season. The landslide displacement again appears to be on an accelerated rise in deformation, which changes from 229.1 mm to 275.3 mm in 2020 and from 278.4 mm to 364.6 mm in 2021.

## 4. Prediction of landslide Displacement

### 4.1. Decomposition of Cumulative Displacement

Cumulative monitoring data (1995 days) between July 2016 and December 2021 were selected for data analysis, yielding 285 weeks of data. Weeks 1–240, 241–265 and 266–285 are classified as the training, validation and test sets, respectively. Four sets of training sets with different time scales were constructed respectively, as shown in Table 1. In order to obtain good landslide displacement prediction results, the displacement needs to be decomposed correctly and the corresponding influential factors must be selected [25,27,46]. The displacement of the landslide due to its own geological conditions is defined as trend term, displacement with periodic variations is used as periodic term, and displacement with other instability influences is used as random term. The $\alpha$ and $\tau$ have a large impact on the decomposition results of the data, after several comparisons, the parameter values that better match the results are finally determined. Hence, the mode numbers $K$, $\alpha$ and $\tau$ are set to 3, 2000 and 0, respectively. Decomposition results of the cumulative displacement data are presented in Figure 4.

**Table 1.** Training sets of the prediction model.

| Model Number | Training Set | Cumulative Data Monitoring | | | |
|:---:|:---:|:---:|:---:|:---:|:---:|
| | | 1–60 Weeks | 61~120 Weeks | 121~180 Weeks | 181~240 Weeks |
| ① | 60 | | | | √ |
| ② | 120 | | | √ | √ |
| ③ | 180 | | √ | √ | √ |
| ④ | 240 | √ | √ | √ | √ |

### 4.2. Decomposition of the Influential Factors

Reservoir water level and rainfall play a considerable role in the landslide deformation process in the studied region [31,38]. The infiltration of precipitation may damage the landslide structure, leading to soil softening and changes in the gravity field of the landslide, as well as surface deformation and displacement of the landslide [47,48]. The dry and wet cycles of landslides caused by the constant regulation of reservoir water level during the

annual cycle also affect the stability of the slopes. Thus, precipitation, reservoir water level and landslide deformation status were chosen as potential influential factors.

**Figure 4.** Decomposition results of the cumulative displacement data.

(1) Displacement factors: average displacement of the current week (D1), average displacement in the last two weeks (D2), and average displacement in the last three weeks (D3).

(2) Precipitation factors: cumulative precipitation of the current week (P1), accumulated precipitation in the current week and last week (P2).

(3) Reservoir water level factors: average reservoir water level in the current week (R1), water level fluctuation data of the current week (R2), water level change data of the current week and the last week (R3).

Considering the possibility that there is no trend term in the influential factor, the decomposition is set to $K = 2$. The component with higher weight and lower frequency is used as periodic influential factor, and the component with smaller weight and higher frequency is used as random influential factor. The decomposition results of some influential factors are presented in Figure 5.

*4.3. Correlation Analysis between the Decomposed Components and Influential Factors*

The reasonable selection of influential factors helps to improve the accuracy of displacement prediction. Therefore, GRDA method was used to determine the relationship between displacement and other influential factors. Notably, the closer the relation value is to 1, the better the relationship between two variables when the discriminant coefficient $\rho$ is 0.5. If the correspondence between the compared sequences is greater than 0.6, we can assume that they are closely related. Table 2 shows the results of GRDA method. It can be observed from Table 2 that there is a significant correlation between the displacement and influential factors.

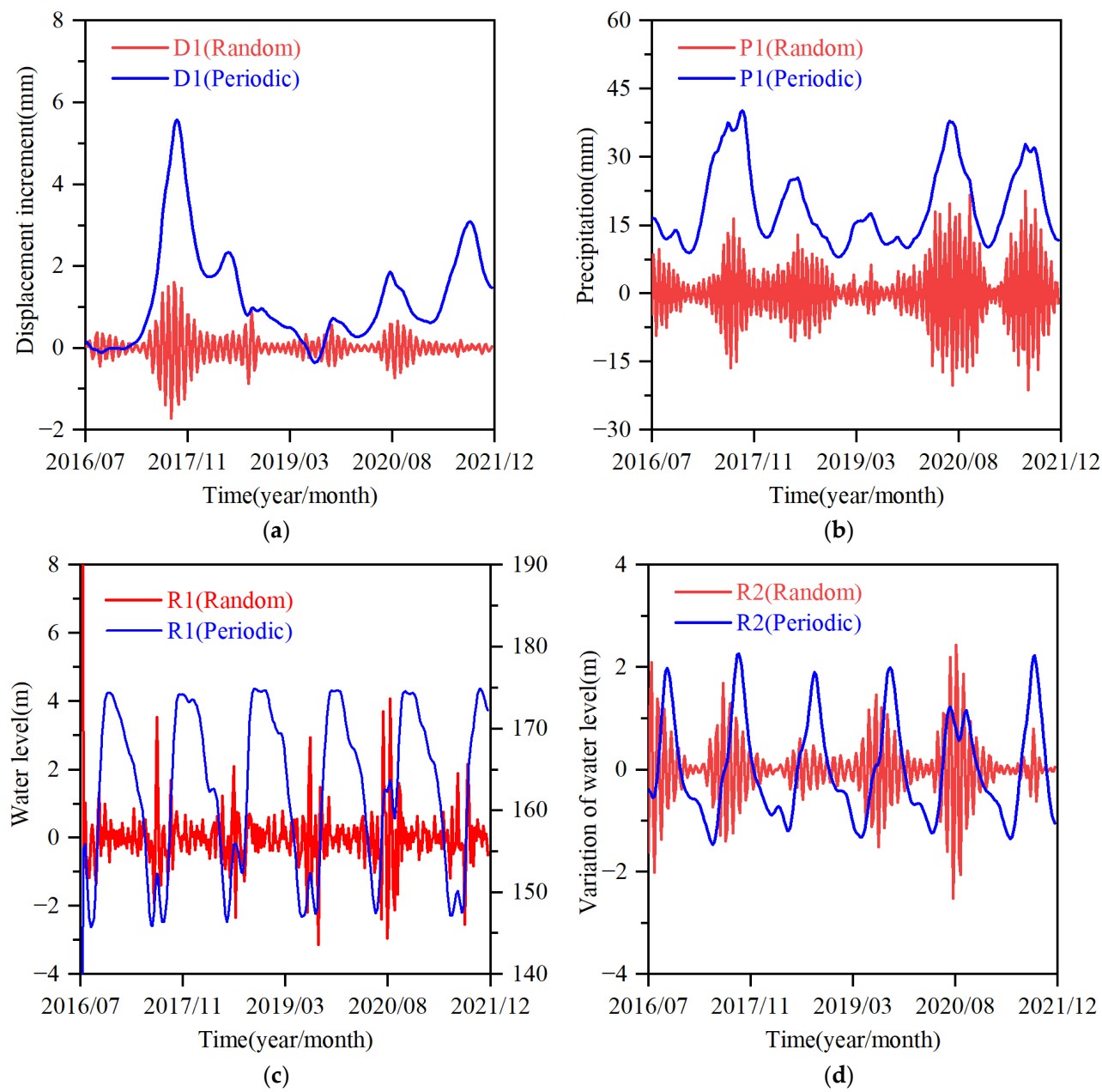

**Figure 5.** Decomposition results of some influential factors. (**a**) D1; (**b**) P1; (**c**) R1; and (**d**) R2.

**Table 2.** GRDA for analyzing the relationship between the displacement and influential factors.

| Model Number | Component | Influential Factors | | | | | | | |
|---|---|---|---|---|---|---|---|---|---|
| | | D1 | D2 | D3 | P1 | P2 | R1 | R2 | R3 |
| ① | Periodic | 0.7685 | 0.7685 | 0.7685 | 0.7679 | 0.7680 | 0.7682 | 0.7323 | 0.7362 |
| | Random | 0.9455 | 0.8222 | 0.9743 | 0.7499 | 0.9697 | 0.7917 | 0.9457 | 0.7807 |
| ② | Periodic | 0.9189 | 0.9190 | 0.9189 | 0.9187 | 0.9188 | 0.9188 | 0.8676 | 0.6441 |
| | Random | 0.9738 | 0.9599 | 0.9706 | 0.9618 | 0.7010 | 0.8039 | 0.9646 | 0.9318 |
| ③ | Periodic | 0.8461 | 0.8464 | 0.8465 | 0.8435 | 0.8437 | 0.8447 | 0.7653 | 0.7800 |
| | Random | 0.9778 | 0.9816 | 0.9592 | 0.9607 | 0.6542 | 0.9239 | 0.9613 | 0.8900 |
| ④ | Periodic | 0.8040 | 0.8040 | 0.8040 | 0.8039 | 0.8039 | 0.8040 | 0.8028 | 0.8036 |
| | Random | 0.8971 | 0.9136 | 0.9187 | 0.7695 | 0.7271 | 0.9338 | 0.7967 | 0.7967 |

### 4.4. Prediction of Trend Displacement

When training and predicting through the GWO-SVR model, the model parameters need to be set first [14,49]. By using several experimental calculations, the relevant parameters in the GWO algorithm are fixed as follows: the number of wolves is 30, the maximum number of iterations is 100, the lower bound of the parameters of $C$, $g$ is 0.01, and the upper bound is 100. The evaluation indicators of trend displacement are listed in Table 3. The values of RMSE and $R^2$ of model ② are better than those of other models, indicating that model ② has an excellent prediction accuracy. The displacement prediction results of model ② are shown in Figure 6.

**Table 3.** Evaluation results of trend displacement.

| Model Number | ① | ② | ③ | ④ |
|---|---|---|---|---|
| MAE (mm) | 0.1212 | 0.1181 | 0.1199 | 0.1294 |
| RMSE (mm) | 0.2426 | 0.2343 | 0.2383 | 0.2666 |
| $R^2$ | 0.9995 | 0.9995 | 0.9995 | 0.9994 |

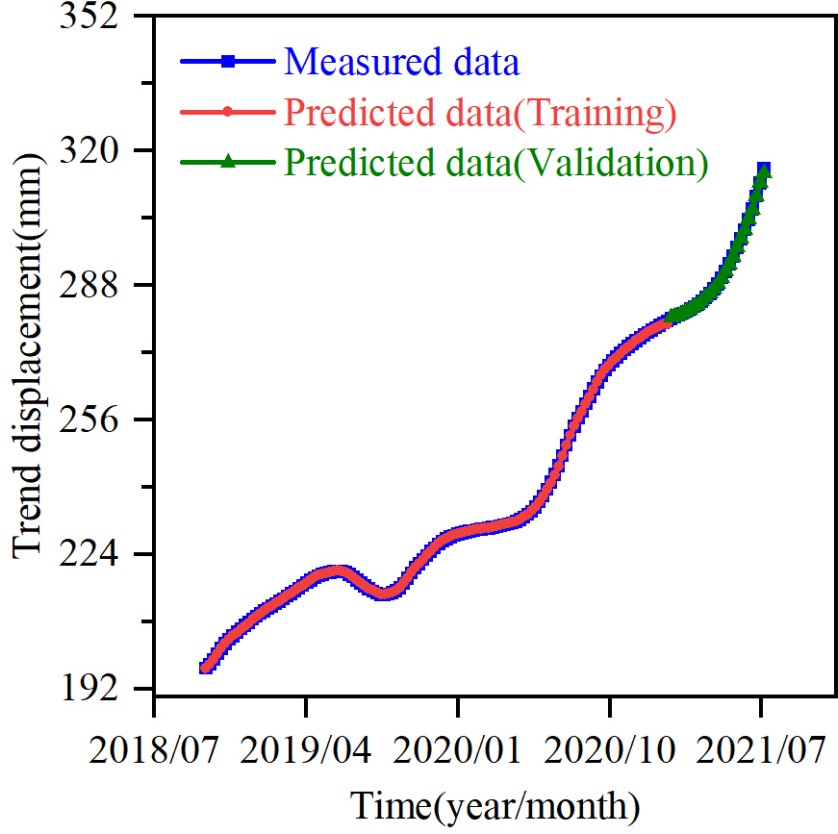

**Figure 6.** Prediction results of trend displacement in both training and validation sets (120 weeks).

### 4.5. Prediction of Periodic Displacement

The evaluation indicators of periodic displacement are listed in Table 4. Notably, model ④ has the highest prediction accuracy with the MAE, RMSE and $R^2$ of 0.0646, 0.0722 and 0.9943, respectively. The displacement prediction results of model ④ are demonstrated in Figure 7. The impacts of precipitation and reservoir water level on landslide displacement are highlighted in the period term. As a result, the period effect from longer monitoring time series may be more obvious. The GWO-SVR model is also well implemented for the displacement prediction of periodic term.

**Table 4.** Evaluation results of periodic displacement.

| Model Number | ① | ② | ③ | ④ |
|---|---|---|---|---|
| MAE (mm) | 0.0925 | 0.0917 | 0.0646 | 0.0646 |
| RMSE (mm) | 0.0944 | 0.0939 | 0.0723 | 0.0722 |
| $R^2$ | 0.9903 | 0.9904 | 0.9943 | 0.9943 |

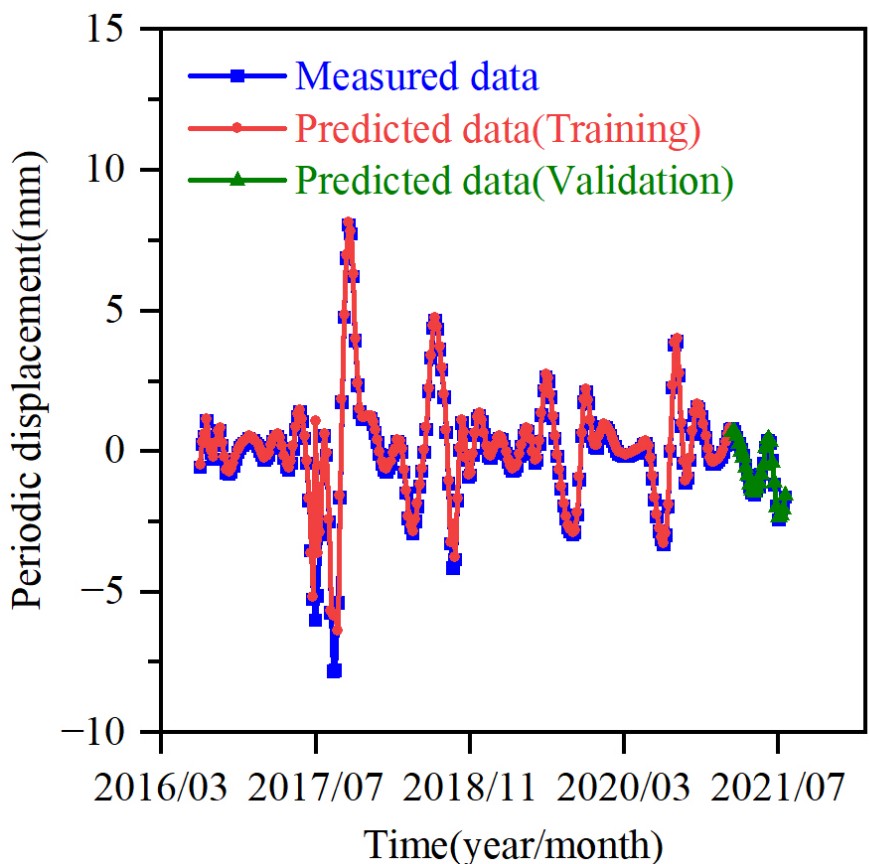

**Figure 7.** Prediction results of periodic displacement in both training and validation sets (240 weeks).

### 4.6. Prediction of Random Displacement

The random component includes the effect of unexpected events such as earthquakes, human activities or other measurement losses, which possesses complex nonlinear character. The evaluation indicators of random displacement are shown in Table 5. Model ④ has the best prediction accuracy, with the MAE, RMSE and $R^2$ of 0.0099, 0.0284 and 0.9849, respectively, at 240 weeks. The displacement prediction results of model ④ are shown in Figure 8. The influence of the random term often includes many unforeseen factors, but the GWO-SVR model achieves the same accuracy of displacement prediction when appropriate random influential factors are selected.

**Table 5.** Evaluation results of random displacement.

| Model Number | ① | ② | ③ | ④ |
|---|---|---|---|---|
| MAE (mm) | 0.1056 | 0.0163 | 0.0267 | 0.0099 |
| RMSE (mm) | 0.1553 | 0.0298 | 0.0359 | 0.0284 |
| $R^2$ | 0.5467 | 0.9833 | 0.9758 | 0.9849 |

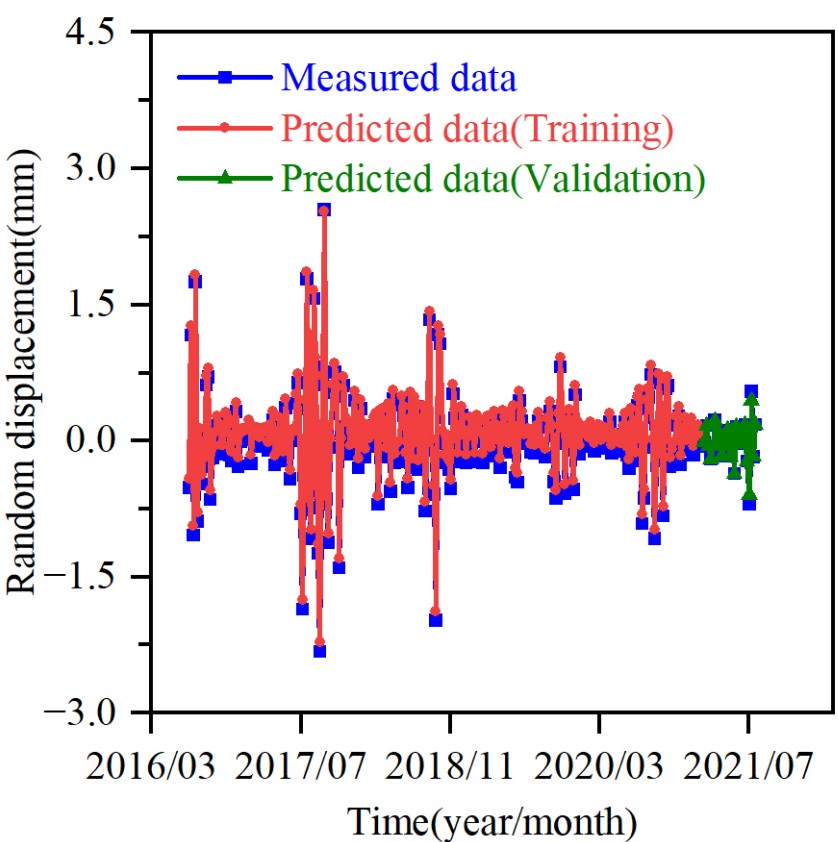

**Figure 8.** Prediction results of random displacement in both training and validation sets (240 weeks).

*4.7. Prediction of Cumulative Displacement*

The three displacement components of trend, period and random were superimposed to obtain the predicted results of cumulative displacement, and the predicted values of each evaluation indicator are listed in Table 6. Notably, the cumulative displacement monitoring values of the displacement prediction models with different training sets are relatively close to the model prediction values, indicating that the displacement prediction models without training sets could basically achieve accurate prediction of landslide displacement. The better prediction performance was achieved when the training set was displaced for 180 weeks, with the MAE, RMSE and $R^2$ of 0.1536, 0.2421 and 0.9994, respectively. The displacement prediction results of model ③ are shown in Figure 9.

**Table 6.** Evaluation results of cumulative displacement.

| Model Number | ① | ② | ③ | ④ | Combination Model |
|---|---|---|---|---|---|
| MAE (mm) | 0.2089 | 0.1633 | 0.1536 | 0.1613 | 0.1489 |
| RMSE (mm) | 0.3054 | 0.2457 | 0.2421 | 0.2695 | 0.2414 |
| $R^2$ | 0.9991 | 0.9994 | 0.9994 | 0.9993 | 0.9994 |

Combining the previous analysis, the trend displacement is best for model ②, while the periodic and random displacements are best for model ④. The optimal combination model could be obtained by combining the three models. The evaluation indicators of the combination prediction model in the validation set are demonstrated in Table 6, with the MAE, RMSE and $R^2$ of 0.1489, 0.2414 and 0.9994, respectively. The results indicate that the optimal combination model has outstanding predictive performance (Figure 10).

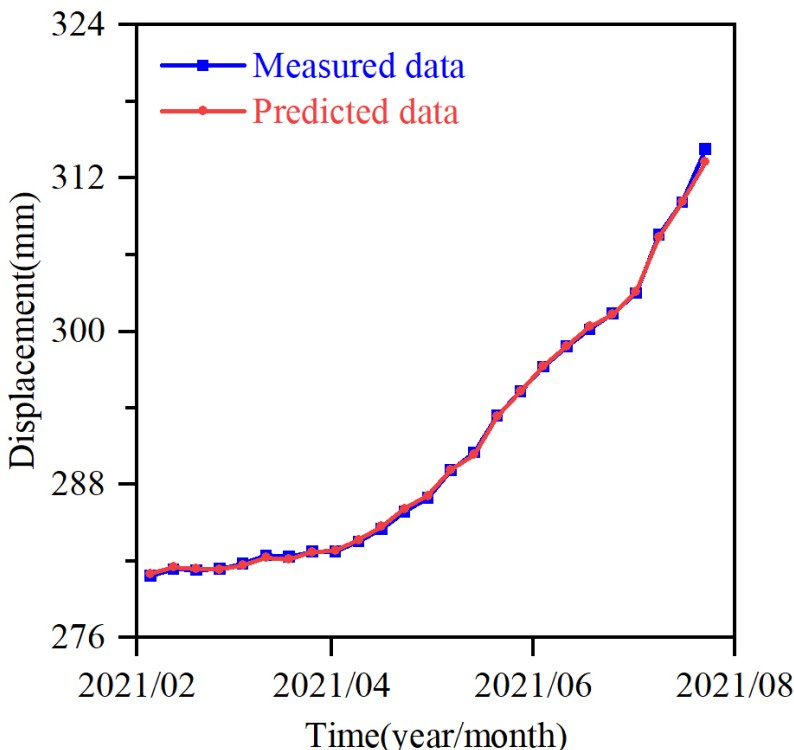

**Figure 9.** Prediction results of cumulative displacement in both training and validation sets (180 weeks).

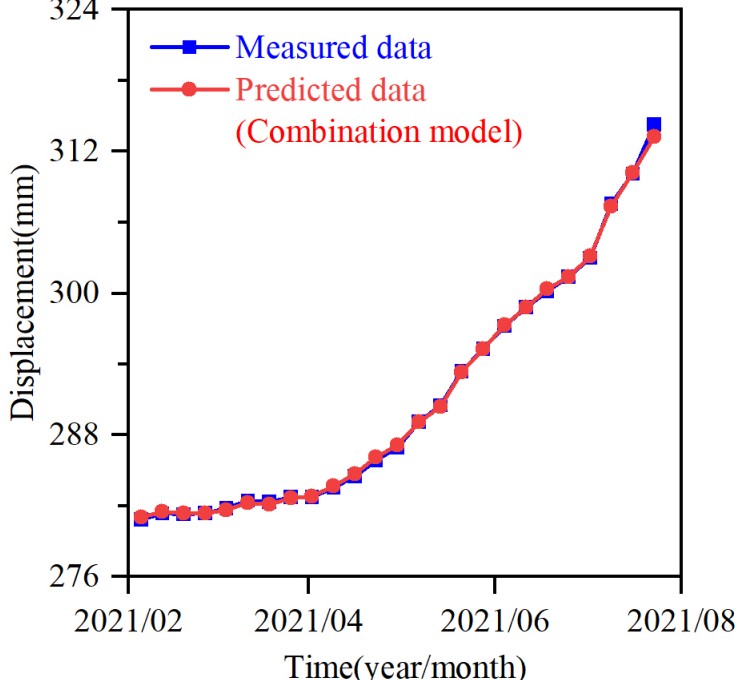

**Figure 10.** Prediction data of the validation set.

The optimal prediction model was evaluated in the test set with the following values: MAE = 0.1993, RMSE = 0.3516, and $R^2$ = 0.9995. The prediction data of the test set are demonstrated in Figure 11.

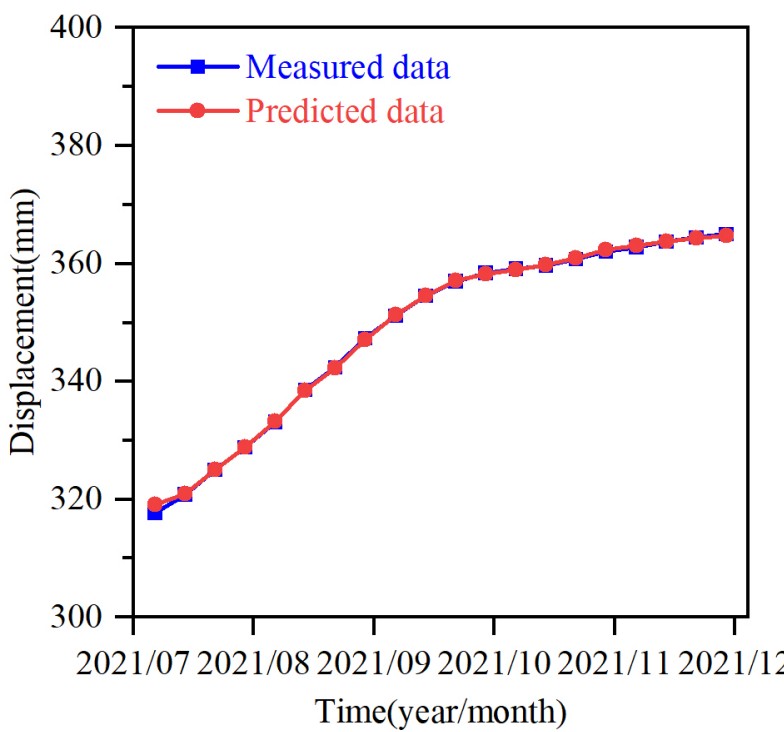

**Figure 11.** Prediction results of the test set.

## 5. Discussion

In general, the causes of landslide deformation in the Three Gorges Reservoir area are influenced by a variety of integrated factors, such as the landslide's geological structure, precipitation and other factors. In this study, seasonal rainfall and adjustment of reservoir water level are found to control the occurrence of step-like displacement through landslide mechanism and monitoring data analysis. The landslide underwent accelerated deformation mainly during the period of heavy precipitation and water level drop, followed by a step-like displacement deformation. Therefore, an effective displacement decomposition method is beneficial for better displacement prediction.

The characteristic components obtained from the decomposition of cumulative displacements by conventional EMD methods are often not fixed [26,27]. Therefore, it is necessary to combine and reconstruct these components to obtain the displacement characteristic components of the landslides. The conventional decomposition method normally produces no less than five characteristic components. The procedures for reconstructing and combining these feature components to obtain the landslide trend, periodic and random components are more complicated and the workload increases significantly, which leads to lower computational efficiency. In this work, landslide displacements are decomposed according to VMD theory, determining the explicit physical meaning of each component. In addition, the VMD theory, which has good adaptive ability, can be used to decompose the displacement based on the actual situation of the landslide [32,33]. Therefore, the trend, period and random displacements of landslides are well extracted in this study. This avoids the phenomenon of over-decomposition or incomplete decomposition of the components caused by the uncertain number of components, especially in the conventional methods such as EMD.

SVR is one of the most typical prediction methods, and landslide displacement can be predicted by optimizing relevant parameters. The reasonable selection of input parameters is helpful to improve the training efficiency of SVR. GWO has the characteristics of fast convergence and high optimization accuracy [25], and it is introduced into SVR for parameter optimization [14,49]. In addition, to improve the prediction accuracy, different factors affecting landslide deformation are analyzed using GRDA, and precipitation and reservoir

level fluctuation data are added to the landslide displacement prediction model as additional influential factors. To further compare the pros and cons of algorithms, prediction analysis is performed with the GWO-SVR and VMD-PSO-SVR models for the test set data, respectively. A comparison of the prediction models is shown in Figure 12. The evaluation indicators of each model are shown in Table 7. The VMD-PSO-SVR and the VMD-GWO-SVR optimization algorithms with VMD decomposition are effective in improving the prediction accuracy compared to that without VMD decomposition. As demonstrated in Figure 12 and Table 7, the RMSE and $R^2$ of VMD-GWO-SVR are greater than those of other models. The VMD-GWO-SVR model has the best prediction performance, which can provide a good decision basis for real-time landslide warning.

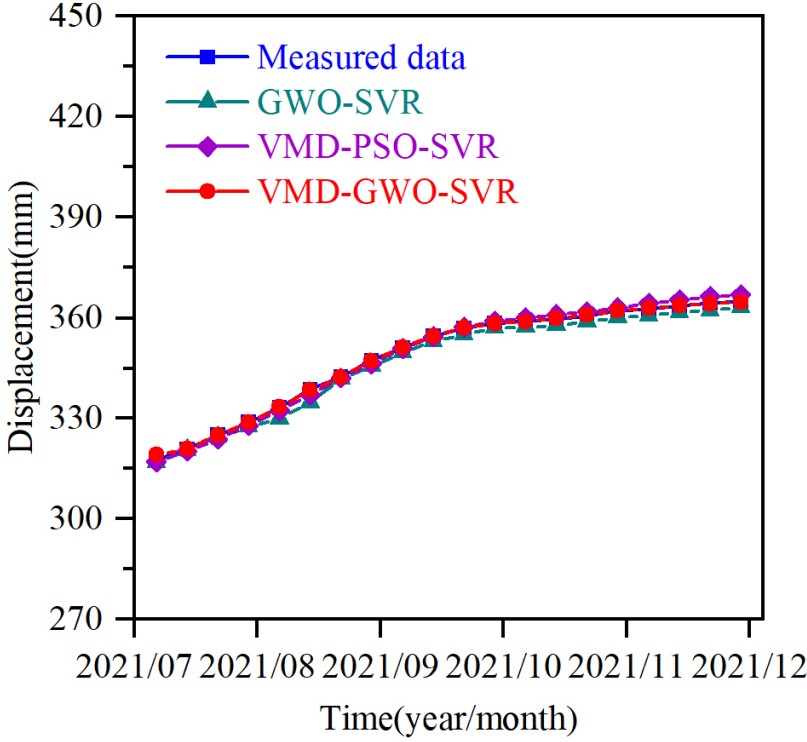

**Figure 12.** Comparative analysis of different displacement prediction models.

**Table 7.** Evaluation index values of the three landslide displacement prediction models.

| Model | GWO-SVR | VMD-PSO-SVR | VMD-GWO-SVR |
|---|---|---|---|
| MAE (mm) | 1.7441 | 1.0106 | 0.1993 |
| RMSE (mm) | 1.9242 | 1.1434 | 0.3516 |
| $R^2$ | 0.9845 | 0.9945 | 0.9995 |

## 6. Conclusions

Given the step-like nonlinear characteristics of landslide displacement, a novel model is proposed to predict landslide displacement based on the VMD theory, GWO algorithm and SVR model. VMD theory is used to effectively extract three components of landslide displacement, each of which represents a distinct characteristic of displacement. The superposition results of the three components are basically consistent with the original displacement, which proves the effectiveness of the displacement decomposition and effectively solves the problem of low computational efficiency. The external factors influencing landslide displacement deformation are accurately identified by GRDA. The experiments on the Shuizhuyuan landslide prove that SVR can effectively solve the nonlinear regression and time series problems. The optimization of SVR model parameters is realized by GWO algorithm and the predictive performance of the above model is confirmed by multiple

validation tests with different time series of training sets. The prediction results of the combination model prove that the model can achieve a high accuracy for predicting landslide displacement. Specifically, under the premise of adequate pre-monitoring information and effective access to long-term landslide monitoring data, the proposed prediction model can be applied for the prediction of landslide displacement in the studied region.

**Author Contributions:** Conceptualization, C.W.; methodology, C.W.; software, C.W.; validation, C.W. and W.G.; formal analysis, C.W.; investigation, W.G.; resources, C.W.; data curation, C.W.; writing—original draft preparation, C.W.; writing—review and editing, C.W.; visualization, C.W. and W.G.; supervision, C.W.; project administration, C.W.; funding acquisition, C.W. All authors have read and agreed to the published version of the manuscript.

**Funding:** This research was funded by the National Key Research and Development Program of China (No. 2018YFC150480502 and No. 2019YFC150960101).

**Institutional Review Board Statement:** Not applicable.

**Informed Consent Statement:** Not applicable.

**Data Availability Statement:** Not applicable.

**Acknowledgments:** The authors would like to express their gratitude to the Three Gorges Reservoir Geological Hazards—Chongqing Wushan Field Scientific Observation and Research Station of the Ministry of Natural Resources for providing the landslide monitoring data of Shuizhuyuan.

**Conflicts of Interest:** The authors declare no conflict of interest.

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
