# Peer review of "Prediction of Landslide Displacement Based on the Variational Mode Decomposition and GWO-SVR Model"

_sustainability, doi:10.3390/su15065470_

Round 1

Reviewer 1 Report

The manuscript (sustainability-2270248) presents a objective of this study is to establish a prediction model for landslide displacement in the Three Gorges Reservoir area using VMD and SVR optimized by GWO-SVR. The model aims to accurately predict the displacement of landslides under the influence of periodic precipitation and reservoir water level changes, thereby reducing the risk of landslide disasters. The study uses the Shuizhuyuan landslide as an example to test the accuracy of the model. It’s an interesting and important topic, considering the last decade many advances in these area.

In relation the manuscript, Introduction, objective are clear, material and methods, results, and conclusion demonstrated a good topic for reader. However, discussion is very small and negligence many relate other studies. Please, considered improve this section. Minor point for my comments and suggestions for Authors:

This research is the background introduction for why these algorithms are important for classification. For example, it could be applied in other areas. New study and other regions for landslide should be applied? In addition, Why are there other models? If so, why were they not applied? It would be better if this background information could be provided.

-Alphabetic order for keywords;

- Add information in Figure 2. Where were these data collected?

-Considering join figures similarly, ex. Figures 8 and 9, etc.

-Discussion sections need improve. 20 lines and nothing references, its not good quality for your revelated results demonstrated in your manuscript. Please, rewrite and improve.

-Check references and add when necessary. In addition, exclude old references.

Best regards,

Author Response

Dear Editors and Reviewers,

We sincerely thank the Editors and the reviewers for your careful and timely review and valuable comments on our manuscript (Manuscript ID: sustainability-2270248). These comments are all valuable and very helpful for revising and improving our paper, as well as the important guiding significance to our researches. We carefully considered and revised the article based on the opinions of reviewers. The revised manuscript of the paper has been revised in review mode and the important revisions are marked in red. The main corrections in the paper and the responds to the reviewer’s comments are as flowing:

Point 1: This research is the background introduction for why these algorithms are important for classification. For example, it could be applied in other areas. New study and other regions for landslide should be applied? In addition, Why are there other models? If so, why were they not applied? It would be better if this background information could be provided.

Response 1: Thank you for providing these insights. (1) The development history of landslide displacement prediction algorithms and different types of prediction algorithms in different periods are briefly described in the paper. Algorithm classification helps us to select a suitable prediction algorithm to predict the displacement of landslides in the Three Gorges Reservoir area. The revised manuscript reformulates this part and points out the shortcomings and drawbacks of some traditional algorithms, and discusses why the integrated coupled model is chosen to carry out the displacement prediction model study. Modified as follows: The empirical model requires creep experiments to validate the prediction model and has limited application scenarios. Mathematical statistical models are better for the prediction of single influential factor, but cannot solve the displacement prediction of multiple influential factors. Nonlinear prediction suffers from slow convergence and is easily trapped in local minima. The integrated coupled model achieves the prediction of landslide displacement from multiple model perspectives and improves the accuracy of displacement prediction. (2) In fact, the displacement prediction algorithm mainly relies on the continuous progress and development of machine learning algorithms, which currently have a wide range of applications in several fields. We drew on relevant references in the selection of algorithms and selected prediction methods applicable to landslide displacement in the Three Gorges Reservoir area. In addition, whether these methods are applicable in other regions deserves further consideration and research. (3) In the Introduction part, we present different displacement decomposition methods, prediction methods and parameter optimization methods according to available references and point out the advantages and disadvantages of the different methods. It is based on these existing research methods that a suitable combined prediction model is selected, and in the Discussion part, a comparative analysis with the existing related research methods is conducted to verify the validity of our proposed model. This part has also been added in the revised manuscript. Modified as follows: The gray wolf optimizer (GWO) has been widely used in combinatorial model optimization problems compared to some existing algorithms (GA, PSO), and in particular, it can greatly improve the efficiency of parameter optimization. Therefore, GWO has also been introduced to realize the optimization of the SVR model parameters [25, 42].

Point 2: Alphabetic order for keywords;

Response 2: Thank you for your suggestion. We have adjusted the order of the keywords.

Point 3: Add information in Figure 2. Where were these data collected?

Response 3: Thanks to the reviewer for your meticulous and careful review. The monitoring data of Shuizhuyuan landslide in the Figure 3 (original Figure 2) are from the Three Gorges Reservoir Geological Hazards — Chongqing Wushan Field Scientific Observation and Research Station of the Ministry of Natural Resources. Relevant information is still under project protection and we add data sources in the acknowledgements.

Point 4: Considering join figures similarly, ex. Figures 8 and 9, etc.

Response 4: Thanks to the reviewer for the suggestion. The prediction data in Figure 9 (original Figure 8) represents the data set with better training model. Figure 10 (original Figure 9) shows a schematic diagram for validating the model with different component combinations. The meaning of the two figures is inconsistent and therefore does not join the two figures. We have also redrawn and modified the two figures.

Point 5: Discussion sections need improve. 20 lines and nothing references, its not good quality for your revelated results demonstrated in your manuscript. Please, rewrite and improve.

Response 5: We think this is an excellent suggestion. We have re-written this part according to the Reviewer’s suggestion. Modified as follows:

In general, the causes of landslide deformation in the Three Gorges Reservoir area contain a variety of integrated factors, such as its own geological structure, precipitation and other factors. In this study, seasonal rainfall and adjustment of reservoir water level are found to control the occurrence of step-like displacement through landslide mechanism and monitoring data analysis. The landslide underwent accelerated deformation mainly during the period of heavy precipitation and water level drop, followed by a step-like displacement deformation. Therefore, an effective displacement decomposition method is beneficial for better displacement prediction.

The characteristic components obtained from the decomposition of cumulative displacements by conventional EMD methods are often not fixed [26, 27]. Therefore, it is necessary to combine and reconstruct these components to obtain the displacement characteristic components of the landslides. The conventional decomposition method normally produces no less than 5 characteristic components. The procedures for reconstructing and combining these feature components to obtain the landslide trend, periodic and random components are more complicated and the workload increases significantly, which leads to lower computational efficiency. In this work, landslide displacements are decomposed according to VMD theory, determining the explicit physical meaning of each component. Besides, the VMD theory, which has good adaptive ability, can be used to decompose the displacement based on the actual situation of the landslide [32, 33]. Therefore, the trend, period and random displacements of land-slides are well extracted in this study. This avoids the phenomenon of over-decomposition or incomplete decomposition of the components caused by the uncertain number of components, especially in the conventional methods such as EMD.

SVR is one of the most typical prediction methods, and landslide displacement can be predicted by optimizing relevant parameters. The reasonable selection of input parameters is helpful to improve the training efficiency of SVR. GWO has the characteristics of fast convergence and high optimization accuracy [25], and it is introduced into SVR for parameter optimization [14, 49]. In addition, to improve the prediction accuracy, different factors affecting landslide deformation are analyzed using GRDA, and precipitation and reservoir level fluctuation data are added to the landslide displacement prediction model as additional influential factors. To further compare the pros and cons of algorithms, prediction analysis is performed with the GWO-SVR and VMD-PSO-SVR models for the test set data, respectively. A comparison of the prediction models is shown in Figure 12. The evaluation indicators of each model are shown in Table 7. The VMD-PSO-SVR and the VMD-GWO-SVR optimization algorithms with VMD decomposition are effective in improving the prediction accuracy compared to that without VMD decomposition. As demonstrated in Figure 12 and Table 7, the RMSE and R2 of VMD-GWO-SVR are greater than those of other models. The VMD-GWO-SVR model has the best prediction performance, which can provide a good decision basis for real-time landslide warning.

Point 6: Check references and add when necessary. In addition, exclude old references.

Response 6: We sincerely appreciate the valuable comments. We have checked the literature carefully and added more references (1. Zhang W, Li H, Han L et al. Slope stability prediction using ensemble learning techniques: a case study in Yunyang County, Chongqing, China. Journal of Rock Mechanics and Geotechnical Engineering 2022, 14, 1089-1099; 2. Phoon K-K, Zhang W. Future of machine learning in geotechnics. Georisk: Assessment and Management of Risk for Engineered Systems and Geohazards 2022, 1-16.) into the Introduction and Discussion parts in the revised manuscript. In addition, we deleted the old references (Yin Y, Wang F, Sun P. Landslide hazards triggered by the 2008 Wenchuan earthquake, Sichuan, China. Landslides 2009, 6, 139-152.).

We tried our best to improve the manuscript and made some changes in the manuscript. These changes will not influence the content and framework of the paper.

We appreciate for Editors/Reviewers’ warm work earnestly, and hope that the correction will meet with approval.

Once again, thank you very much for your comments and suggestions.

Best regards,

Chenhui Wang

E-mail: wangchenhui@mail.cgs.gov.cn

Reviewer 2 Report

In this study, the authors proposed a landslide displacement prediction method based on VMD and GWO-SVR. This topic is of great significance for landslide early warning and risk reduction. This paper is well prepared and structured. Both methodology and results are better described. In my opinion, before the publication of this manuscript, some minor revisions are needed.

1. Back propagation neural network can be represented by the abbreviation BPNN, so BP network is not necessary in the text and can be replaced by BPNN.

2. The introduction of the optimization algorithm should complement the advantages of the optimization algorithm.

3. The "influence factors" in Figure 1 should be revised to " influential factors ", and the expression of influence factors in the text should be unified.

4. The paper describes the reason for setting K in the VMD algorithm and how the other two parameters are set.

5. The engineering geological profile of Shuizhuyuan landslide should be added in 3.1.

6. What is the reason for selecting SZY-03 monitoring point.

7. Recent relevant references should be included, such as “Zhang Wengang, Li Hongrui, Han Liang, Chen Longlong, Wang Lin. Prediction of slope stability using ensemble learning techniques: a case study in Yunyang County, Chongqing, China. Journal of Rock Mechanics and Geotechnical Engineering. 14 (2022) 1089-1099.” “Kok-Kwang Phoon & Wengang Zhang (2022): Future of machine learning in geotechnics, Georisk: Assessment and Management of Risk for Engineered Systems and Geohazards, DOI: 10.1080/17499518.2022.2087884”.

Author Response

Dear Editors and Reviewers,

Thank you very much and the reviewers for your careful and timely review and valuable comments on our manuscript (Manuscript ID: sustainability-2270248). Those comments are all valuable and very helpful for revising and improving our paper, as well as the important guiding significance to our researches. We carefully considered and revised the article based on the opinions of reviewers. The revised manuscript of the paper has been revised in review mode and the important revisions are marked in red. The main corrections in the paper and the responds to the reviewer’s comments are as flowing:

Point 1: Back propagation neural network can be represented by the abbreviation BPNN, so BP network is not necessary in the text and can be replaced by BPNN.

Response 1: Thanks to the reviewer for the suggestion. The back propagation neural network (BPNN) has been introduced in the paper. We have carefully checked the full text and modified the BP neural network to BPNN.

Point 2: The introduction of the optimization algorithm should complement the advantages of the optimization algorithm.

Response 2: Thanks to the reviewer for the suggestion. As the reviewer mentioned the optimization algorithm has many advantages. The advantages of the optimization algorithm are simple structure, faster convergence and fast access to optimization parameters. We have added to the revised manuscript. Modify as follows: The gray wolf optimizer (GWO) has been widely used in combinatorial model optimization problems compared to some existing algorithms (GA, PSO), and in particular, it can greatly improve the efficiency of parameter optimization. Therefore, GWO algorithm has also been introduced to realize the optimization of the SVR model parameters [25, 42].

Point 3: The "influence factors" in Figure 1 should be revised to " influential factors ", and the expression of influence factors in the text should be unified.

Response 3: Thanks to the reviewer for their meticulous and careful review. This is indeed a mistake in the preparation of our paper. We have replaced the Figure 1 and checked the full text.

Point 4: The paper describes the reason for setting K in the VMD algorithm and how the other two parameters are set.

Response 4: Thanks to the reviewer for the suggestion. During the VMD decomposition, we executed several comparison experiments. Based on the results of the comparative analysis of the experiments, the α was set to 2000 and the τ was set to 0. During the experiment, the α and τ have a large impact on the decomposition results of the data, especially in the trend term, the period term and the random term of the observed displacement. The results of different parameter decompositions varied greatly. After several comparisons, the parameter values that better match the results were finally determined. Modify as follows: The displacement of landslide due to its own geological conditions is defined as trend term; displacement with periodic variations is used as periodic term; and displacement with other instability influences is used as random term. The α and τ have a large impact on the decomposition results of the data, after several comparisons, the parameter values that better match the results are finally determined. Hence, the mode number K, α and τ are set to 3, 2000 and 0, respectively.

Point 5: The engineering geological profile of Shuizhuyuan landslide should be added in 3.1.

Response 5: Thanks to the reviewer for the suggestion. The engineering geological profile of Shuizhuyuan landslide has been added to the revised manuscript in 3.1.

Point 6: What is the reason for selecting SZY-03 monitoring point.

Response 6: The selection of the landslide displacement value needs to fully consider the influence of rainfall and reservoir water level on the landslide displacement monitoring data. In Shuizhuyuan landslide, the monitoring point SZY-03 is located in the wading part of the front edge of the landslide and its deformation data is the most obvious. In order to study the applicability of the prediction model, we preferentially select the monitoring data of the SZY-03 monitoring point with obvious deformation for landslide displacement prediction. Modify as follows: Combined with the landslide deformation trend, the monitoring point SZY-03 is located in the front edge of the landslide and its data is the most obvious. Therefore, SZY-03 monitoring point among them is selected for data analysis, as shown in Figure 3.

Point 7: Recent relevant references should be included, such as “Zhang Wengang, Li Hongrui, Han Liang, Chen Longlong, Wang Lin. Prediction of slope stability using ensemble learning techniques: a case study in Yunyang County, Chongqing, China. Journal of Rock Mechanics and Geotechnical Engineering. 14 (2022) 1089-1099.” “Kok-Kwang Phoon & Wengang Zhang (2022): Future of machine learning in geotechnics, Georisk: Assessment and Management of Risk for Engineered Systems and Geohazards, DOI: 10.1080/17499518.2022.2087884”.

Response 7: Thanks to the reviewer for the suggestion. It is really true as Reviewer suggested that recent relevant references should be included. The papers mentioned have been added to the References.

  1. Zhang W, Li H, Han L et al. Slope stability prediction using ensemble learning techniques: a case study in Yunyang County, Chongqing, China. Journal of Rock Mechanics and Geotechnical Engineering 2022, 14, 1089-1099.
  2. Phoon K-K, Zhang W. Future of machine learning in geotechnics. Georisk: Assessment and Management of Risk for Engineered Systems and Geohazards 2022, 1-16.

We tried our best to improve the manuscript and made some changes in the manuscript. These changes will not influence the content and framework of the paper.

We appreciate for Editors/Reviewers’ warm work earnestly, and hope that the correction will meet with approval.

Once again, thank you very much for your comments and suggestions.

Best regards,

Chenhui Wang

E-mail: wangchenhui@mail.cgs.gov.cn

Reviewer 3 Report

This paper predicts landslide displacement with step-like deformation characteristics in the Three Gorges Reservoir area. The paper combines VMD, GWO and SVR models to form a VMD-GWO-SVR prediction model. The model achieves effective prediction of landslide displacement in Shuizhuyuan with high prediction accuracy. The language and structure of the paper is good. The work is novel and the approach is practical. I recommend that the paper be accepted with a minor revision. Some suggestions are made as follows:

(1) The engineering geological profile of Shuizhuyuan landslide should be given in the paper.

(2) Landslide mechanism analysis helps to better carry out landslide displacement prediction, and I suggest to supplement the analysis of landslide mechanism in Shuizhuyuan.

(3) The Shuizhuyuan landslide with step-like displacement is the focus of this paper. However, the reasons of this particular deformation have not been elucidated.

(4) The innovation of this research should be illustared in the discussion section. Further, this research should be compared to other relavant researches.

(5) The conclusion needs further induction and summary.

(6) Some figures are not clear enough.

Author Response

Dear Editors and Reviewers,

Thank you very much and the reviewers for your careful and timely review and valuable comments on our manuscript (Manuscript ID: sustainability-2270248). Those comments are all valuable and very helpful for revising and improving our paper, as well as the important guiding significance to our researches. We carefully considered and revised the article based on the opinions of reviewers. The revised manuscript of the paper has been revised in review mode and the important revisions are marked in red. The main corrections in the paper and the responds to the reviewer’s comments are as flowing:

Point 1: The engineering geological profile of Shuizhuyuan landslide should be given in the paper.

Response 1: Thanks to the reviewer for the suggestion. The engineering geological profile of Shuizhuyuan landslide has been added to the revised manuscript in 3.1.

Point 2: Landslide mechanism analysis helps to better carry out landslide displacement prediction, and I suggest to supplement the analysis of landslide mechanism in Shuizhuyuan.

Response 2: Thanks to the reviewer for the suggestion. The revised manuscript adds an in-depth analysis of the landslide mechanism section of Shuizhuyuan landslide. Modify as follows: The landslide material is mainly composed of Quaternary landslide accumulation gravelly soil layer with loose structure, which is easy to be softened by rainfall infiltration and causes sliding deformation. The contact surface between the surface loose soil and bedrock is the sliding surface of the landslide. Rainfall infiltration and reservoir water level change can cause the hydraulic gradient of groundwater in the landslide body to increase, and the pore water pressure to increase, which will increase the sliding force of the landslide.

Point 3: The Shuizhuyuan landslide with step-like displacement is the focus of this paper. However, the reasons of this particular deformation have not been elucidated.

Response 3: Thanks to the reviewer for your meticulous and careful review. The reasons for this special deformation have been elucidated in 3.2. Modify as follows: The landslide in the Three Gorges Reservoir area is affected by the periodic rise and fall of the reservoir water level as well as the annual rainy season. During the adjustment of the reservoir water level and strong precipitation, the displacement deformation is faster, while the deformation is slow during the non-rainy season or the relatively stable period of the reservoir water level. Therefore, the displacement has relatively obvious step-like characteristics (Figure 3).

Point 4: The innovation of this research should be illustrated in the discussion section. Further, this research should be compared to other relevant researches.

Response 4: Thanks to the reviewer for the suggestion. We have already addressed the innovative points of the algorithm in the discussion section. In addition, the superiority of the proposed model is verified by comparing it with other research methods. Modify as follows:

In general, the causes of landslide deformation in the Three Gorges Reservoir area contain a variety of integrated factors, such as its own geological structure, precipitation and other factors. In this study, seasonal rainfall and adjustment of reservoir water level are found to control the occurrence of step-like displacement through landslide mechanism and monitoring data analysis. The landslide underwent accelerated deformation mainly during the period of heavy precipitation and water level drop, followed by a step-like displacement deformation. Therefore, an effective displacement decomposition method is beneficial for better displacement prediction.

The characteristic components obtained from the decomposition of cumulative displacements by conventional EMD methods are often not fixed [26, 27]. Therefore, it is necessary to combine and reconstruct these components to obtain the displacement characteristic components of the landslides. The conventional decomposition method normally produces no less than 5 characteristic components. The procedures for reconstructing and combining these feature components to obtain the landslide trend, periodic and random components are more complicated and the workload increases significantly, which leads to lower computational efficiency. In this work, landslide displacements are decomposed according to VMD theory, determining the explicit physical meaning of each component. Besides, the VMD theory, which has good adaptive ability, can be used to decompose the displacement based on the actual situation of the landslide [32, 33]. Therefore, the trend, period and random displacements of land-slides are well extracted in this study. This avoids the phenomenon of over-decomposition or incomplete decomposition of the components caused by the uncertain number of components, especially in the conventional methods such as EMD.

SVR is one of the most typical prediction methods, and landslide displacement can be predicted by optimizing relevant parameters. The reasonable selection of input parameters is helpful to improve the training efficiency of SVR. GWO has the characteristics of fast convergence and high optimization accuracy [25], and it is introduced into SVR for parameter optimization [14, 49]. In addition, to improve the prediction accuracy, different factors affecting landslide deformation are analyzed using GRDA, and precipitation and reservoir level fluctuation data are added to the landslide displacement prediction model as additional influential factors. To further compare the pros and cons of algorithms, prediction analysis is performed with the GWO-SVR and VMD-PSO-SVR models for the test set data, respectively. A comparison of the prediction models is shown in Figure 12. The evaluation indicators of each model are shown in Table 7. The VMD-PSO-SVR and the VMD-GWO-SVR optimization algorithms with VMD decomposition are effective in improving the prediction accuracy compared to that without VMD decomposition. As demonstrated in Figure 12 and Table 7, the RMSE and R2 of VMD-GWO-SVR are greater than those of other models. The VMD-GWO-SVR model has the best prediction performance, which can provide a good decision basis for real-time landslide warning.

Point 5: The conclusion needs further induction and summary.

Response 5: Thanks to the reviewer for the suggestion. We have summarized the conclusion again. Modify as follows: Given the step-like nonlinear characteristics of landslide displacement in the Three Gorges Reservoir area, a novel model is proposed to predict landslide displacement based on the VMD theory, GWO algorithm and SVR model. In the landslide displacement prediction, we consider the internal geological conditions and external environmental influential factors. VMD theory is used to effectively extract three components of landslide displacement, each of which represents a distinct characteristic of displacement. The superposition results of the three components are basically consistent with the original displacement, which proves the effectiveness of the displacement decomposition and effectively solves the problem of low computational efficiency. The external factors influencing landslide displacement deformation are accurately identified by GRDA. The experiments in Shuizhuyuan landslide prove that SVR can effectively solve the nonlinear regression and time series problems. The optimization of SVR model parameters is realized by GWO algorithm and the predictive performance of the above model is confirmed by multiple validation tests with different time series of training sets. The prediction results of the combination model prove that the model can achieve a high accuracy for predicting landslide displacement. Specifically, under the premise of adequate pre-monitoring information and effective access to long-term landslide monitoring data, the proposed prediction model can be applied for the prediction of landslide displacement in the studied region.

Point 6: Some figures are not clear enough.

Response 6: Thank you for the suggestion. We have re-replaced some unclear figures in the revised manuscript (Figure 9, Figure 10, Figure 11).

We tried our best to improve the manuscript and made some changes in the manuscript. These changes will not influence the content and framework of the paper.

We appreciate for Editors/Reviewers’ warm work earnestly, and hope that the correction will meet with approval.

Once again, thank you very much for your comments and suggestions.

Best regards,

Chenhui Wang

E-mail: wangchenhui@mail.cgs.gov.cn
